# Insights on aquatic microbiome of the Indian Sundarbans mangrove areas

**Paltu Kumar Dhal**[1]*, **Germán A. Kopprio**[2], **Astrid Gärdes**[2]

**1** Department of Life Science and Biotechnology, Jadavpur University, Kolkata, India, **2** Department of Biogeochemistry and Geology, Tropical Marine Microbiology, Leibniz Center for Tropical Marine Research, Bremen, Germany

* paltuk.dhal@jadavpuruniversity.in

## Abstract

### Background

Anthropogenic perturbations have strong impact on water quality and ecological health of mangrove areas of Indian Sundarbans. Diversity in microbial community composition is important causes for maintaining the health of the mangrove ecosystem. However, microbial communities of estuarine water in Indian Sundarbans mangrove areas and environmental determinants that contribute to those communities were seldom studied.

### Methods

Nevertheless, this study attempted first to report bacterial and archaeal communities simultaneously in the water from Matla River and Thakuran River of Maipith coastal areas more accurately using 16S rRNA gene-based amplicon approaches. Attempt also been made to assess the capability of the environmental parameters for explaining the variation in microbial community composition.

### Results

Our investigation indicates the dominancy of halophilic marine bacteria from families *Flavobacteriaceae* and OM1 clade in the water with lower nutrient load collected from costal regions of a small Island of Sundarban Mangroves (ISM). At higher eutrophic conditions, changes in bacterial communities in Open Marine Water (OMW) were detected, where some of the marine hydrocarbons degrading bacteria under families *Oceanospirillaceae* and *Spongiibacteraceae* were dominated. While most abundant bacterial family *Rhodobacteracea* almost equally (18% of the total community) dominated in both sites. Minor variation in the composition of archaeal community was also observed between OMW and ISM. Redundancy analysis indicates a combination of total nitrogen and dissolved inorganic nutrients for OMW and for ISM, salinity and total nitrogen was responsible for explaining the changes in their respective microbial community composition.

**Data Availability Statement:** All relevant data are within the manuscript and its Supporting Information files

**Funding:** This work was part of the Joint NAM S&T Centre – ZMT Bremen Fellowship in Tropical

Coastal Marine Research that was supported by the Leibniz Zentrumfür Marine Tropenforschung (ZMT), Bremen, Germany

**Competing interests:** The authors have declared that no competing interests exist.

## Conclusions

Our study contributes the first conclusive overview on how do multiple environmental/ anthropogenic stressors (salinity, pollution, eutrophication, land-use) affect the Sundarban estuary water and consequently the microbial communities in concert. However, systematic approaches with more samples for evaluating the effect of environmental pollutions on mangrove microbial communities are recommended.

## 1. Introduction

Sundarbans, the largest mangrove forest of the world, is situated in the joint delta of Ganges, Brahmaputra and Meghna rivers at Bay of Bengal [1,2]. This UNESCO World Heritage site comprises the Indian state of West Bengal and southwest Bangladesh [3]. Livelihood and well being of millions of people live in and around of Sundarbans, depends on its status and ecological services. Despite its high ecological and economical values, Sundarbans is seriously threatened by different anthropogenic activities. Since the early 19[th] century, landscapes of Sundarbans have also been changing due to saline and freshwater imbalances. Water quality of this ecosystem is largely affected by sewage pollutant originated from industries located upstream and urban areas of West Bengal. Sewage entering into coastal water contains diverse chemical and microbiological pollutants and a wide variety of organic and inorganic wastes [4,5], driving changes on its ecological and physiological health.

Microbial communities of mangroves are responsible for nutrient cycling and play a vital role in productivity, conservation and rehabilitation of mangrove ecosystems [6]. Therefore, understanding their responses to environmental changes is essential to predict changes in service-provisioning [7]. Several recent studies described the microbial community compositions of surface sediments and water of Indian Sundarban mangrove areas. Surface sediments this area dominanted with *Deltaproteobacteria* followed by *Gammaproteobacteria*, *Alphaproteobacteria*, *Betaproteobacteria*, *and Epsilonproteobacteria* under phylum *Proteobacteria*. Abundant bacterial orders are *Desulfobacterales*, *Desulfuromonadales*, *Myxococcales*, *and Bdellovibrionales*. [8–10]. While bacterioplankton communities in the water of this region were found to be abundant with *Gammaproteobacteria* and *Alphaproteobacteria*. At the family level dominancy of *Hyphomicrobiaceae*, *Rhodobacteraceae*, *Pseudomonadaceae*, *Erythrobacteraceae*, *Kordiimonadaceae*, *Hyphomonadaceae*, and *Ruminococcaceae* were observed [11–13]. However, sampling locations of those studies on microbial communities in the Indian Sundarban mangrove water mainly restricted near to an island (Sagar Island) and other estuary of Mooriganga, Thakuran, Matla and Harinbhanga, therefore the major conclusions of these studies were made based on a limited number of samples. Moreover, not much effort has been made to investigate the archaeal community of this region except single report by [3] on surface and subsurface sediments of Indian Sundarban mangrove forest. Moreover, the above studies have rarely analyzed the bacterial and archaeal community structures of the same samples at the same time. Therefore, our knowledge on those communities as well as information on how they are controlled by environmental parameters is limited. In order to assess the microbial communities of marine ecosystem via high-throughput sequencing of amplified 16S rRNA genes with high resolution and fidelity, it is extremely important to select the proper primer set that can't underestimated or overestimated any common marine taxa [14]. However, this will be the first attempt to visualize the accurate and well-resolved picture of bacterial and archaeal communities simultaneously of marine water in Sundarbans mangrove using next-

generation amplicon sequencing of the 16S rRNA gene using recently developed 515F-Y/926R primers that target V4-V5 region of 16S rRNA gene. We also tried to explore the environmental determinants that contribute to the variation of their microbial communities. This study will provide baseline knowledge on microbial ecology of the World Heritage site and serve as a baseline for monitoring programs and predicting changes at impacted sites.

## 2. Material and methods

### 2.1 Ethics statement

The water samples were brought to ZMT, Germany for its environmental parameters and nutrients measurements. The extracted mutagenic DNA were also carried to ZMT, Germany for sequencing and further analysis. In this reference, The National Biodiversity Authority (NBA), Government of India is well aware of this activity (NBA/Tech Appl/9/Form B/52/17/17-18/2985) that the above mentioned samples are being used only for research purpose. Further no permits were prerequisite for the defined field studies, which complied with all relevant regulation neither the studied locations are not privately owned.

### 2.2 Study sites and sample collection

In the present study sampling was conducted in the Sundarbans mangrove ecosystem that shared between India and Bangladesh and lies in the Ganga-Brahmaputra-Meghna (GBM) delta. This mangrove ecosystem contains over 102 islands with a network complex of many rivers, rivulets, and creeks [15]. Sampling was carried out at two different locations on Thakuran River—Matla River estuarine complex of Maipith coastal areas in the Indian Sundarbans mangroves during March 2017 during low tide (Fig 1). They are designated as Island of Sundarban Mangroves (ISM) and Open Marine Water (OMW). ISM is an uninhabited small island with lesser anthropogenic disturbance situated in Thakuran and Matla river complex in low-lying coastal plain. This river has no reports for perennial fresh water source [15]. Water from three different coastal regions of this island is selected for sampling. OMW is an open marine site around same regions, which is supposedly continually influenced by the wastewaters from upstream regions of Matla River. Three independent replicated water samples (1 L) from each of three different sampling sites of both ISM and OMW were collected in sterile containers and immediately stored at a chilled box until further laboratory analysis.

### 2.3 Environmental parameters and nutrients analysis

Physiochemical parameters (salinity and pH) of all collected samples were measured using Eureka 2 Manta multiprobe (Eureka Environmental Engineering, Texas, USA). Total 50 mL of each sample was filtered through a 0.7 μm syringe filter and poisoned with 200 μL of 3.5 g/100 mL $HgCl_2$ solution for nutrient analysis. Now each of the treated samples was filtered through a 0.7 μm pore size GF/F filter (GE Healthcare Bio-Sciences, Pittsburgh, PA, USA) for DOC and nutrients measurements. The total nitrogen (TN) content of each the samples were measured using the EuroVector EA 3000 elemental analyzer. For measurements of DOC, the filtered samples were acidified with concentrated HCl (pH <2) and analyzed by high-temperature catalytic oxidation method method using a TOC-VCPN analyzer (Shimadzu, Mandel, Canada). Seawater standards (Hansell laboratory, RSMAS University Miami, USA) were used for calibration and quality control, and ultrapure water as a blank. The Dissolved inorganic nutrients that includes combined nitrate and nitrite (NOx), phosphate ($PO_4^{3-}$-P), and silicate ($Si(OH)^{4-}$) were analyzed using a continuous flow analyzer (Flowsys by Unity Scientific,

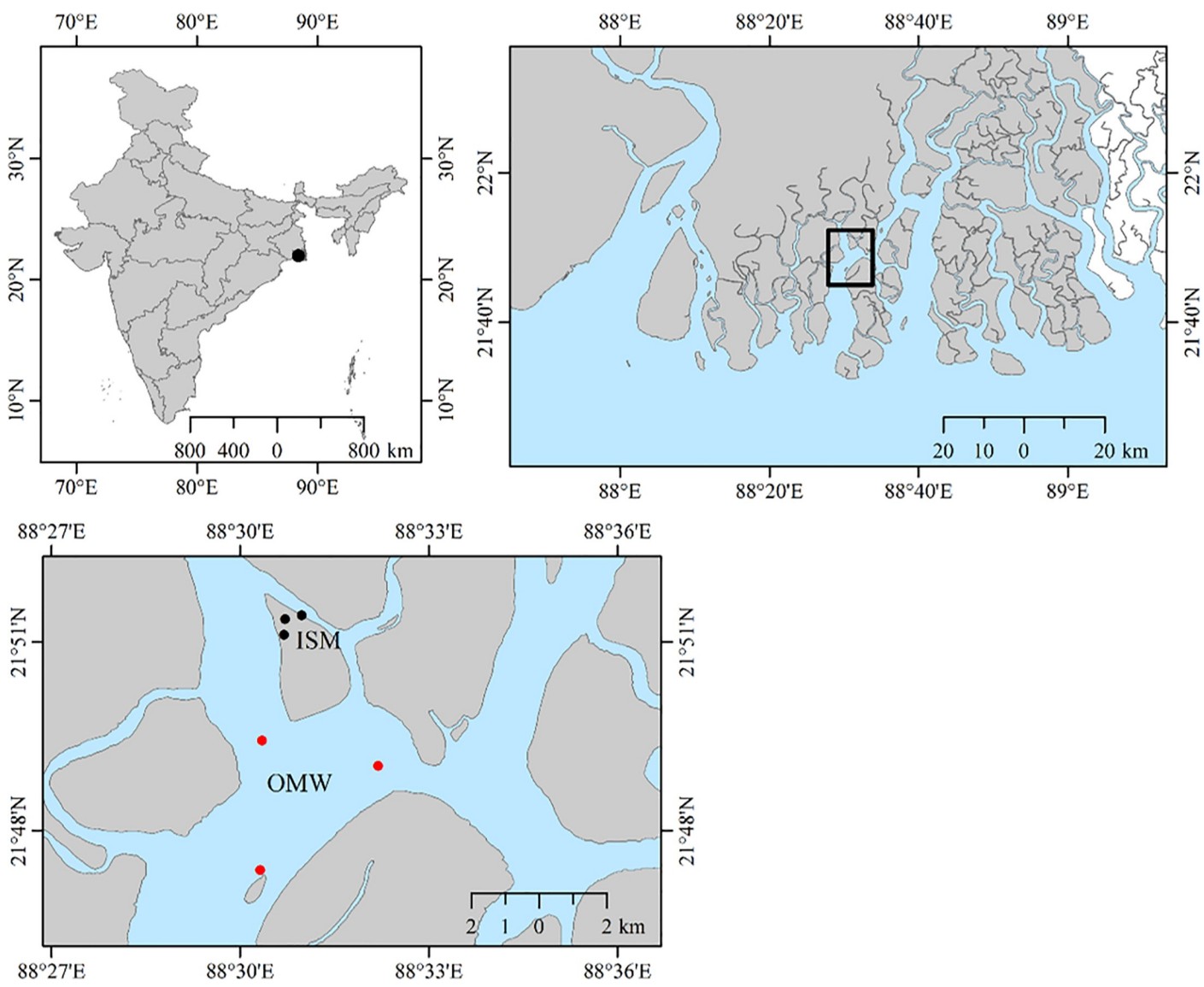

**Fig 1. Map of the sampling area: Water samples were collected from three stations (KL 1, KP 1 and TH1) of a small Island are named as ISM and open marine water samples named as OWM (TH 2, KL 2 and BL 1).** Three biological replicates from each of the six stations; therefore, total eighteen (18) samples (nine from ISM and another nine from OMW) were collected for this investigation. The GPS data of sampling sites were collecting during sampling and compiled together in an ArcGIS10.3 software environments and finally map has been prepared using open source database of GADM (https://gadm.org/).

Brookfield, USA) and detected spectrophotometrically as a colored complex [16] (https://doi.pangaea.de/10.1594/PANGAEA.889699).

## 2.4 DNA extraction, PCR amplification, and Illumina MiSeq sequencing

From each site, water (1 L) was filtered (0.2 μm) and DNA was extracted using the Power-Water® DNA Isolation Kit according to the manufacturer's instructions (MoBio Laboratories Inc., Carlsbad, CA, USA). DNA concentrations and purity were measured spectrophotometrically. Presence of bacterial and archebacterial 16S rRNA gene was in the extracted metagenome was verified following previous method [17]. In order to classify taxonomically both bacterial and archaeal community structure simultaneously, sequencing of V4–V5

hypervariable regions of 16S rRNA gene were generated using primers 515F-Y (5′–GTGYCA GCMGCCGCGGTAA–3′) and 926R (5′–CCGYCAATTYMTTTRAGTTT–3′) [14] on the Illumina MiSeq platform (CeBiTec Bielefeld, Germany), in a 2 × 300 bp paired-end run.

## 2.5 High throughput sequencing data processing

Primer sequences were removed using *cutadapt* from the raw paired-end reads [18]. The primer-trimmed sequences are available on Sequence Read Archive (SRA) (accession no. SRP144285). Sequences were quality trimmed with *trimmomatic* v0.32 [19] using a sliding window of 4 bases and a minimum average quality of 15, and merged with PEAR v0.9.5 [20]. Quality-filtered sequences were clustered into OTUs with *swarm* algorithm using default parameters [21]. One single representative sequence per OTU was taxonomically classified with SINA (SILVA Incremental Aligner; v1.2.11; Silva reference database release 132) at a minimum alignment similarity of 0.9, and a last common ancestor consensus of 0.7 [22]. OTUs that were unclassified on the domain level and those matching to chloroplast and mitochondrial sequences were excluded from the analysis using well-standardized r script [23]. The final OTU tables are accessible at (https://doi.pangaea.de/10.1594/PANGAEA.890757).

## 2.6 Statistical analysis

Principal component analysis (PCA) was performed to cluster the sampling sites based on their environmental parameters. Differences in environmental parameters among ISM and OMW were assessed using general linear mixed models (GLMM) with sampling station as a random factor [24].

Alpha-Diversity indices were calculated to assess richness and evenness of the microbial communities [25] in the studied samples, based on repeated random subsampling of the amplicon data sets after randomly rarefying the data set to the minimum library size (50517 sequences). Significant differences in alpha-diversity indices between the studied stations were determined by using the non-parametric Kruskal test followed by p-value adjusted Wilcoxon tests [26].

To assess the differences in community structure between two sampling sites (beta-diversity), Bray–Curtis dissimilarities were calculated using the relative OTU abundances and also non-metric multidimensional scaling (NMDS) plot was produced. Analysis of similarity (ANOSIM) was calculated to assess the separation of bacterial communities between the two sites. P-values of all multiple pairwise comparisons were adjusted using the false discovery rate (fdr) correction method by [27]. In order to evaluate the environmental parameters as drivers of the variations in community compositions, redundancy analysis (RDA) was used with centered log ratio (clr)-transformed sequence counts using the R function aldex.clr of the ALDEx2 package via median values of 128 Monte-Carlo instances [28]. To compare the explanatory power of all measured environmental parameters, additional RDA models were constructed with environmental parameters as predictors. Forward model selection was used after checking for variance inflation to determine which of parameters would be included in the RDA models. When more than one parameter was included, pure effects were also tested accounting for the variation explained by the other factors in the model. Collinearity among predictors was determined via Variance inflation factors (VIFs) of the individual parameters. All of the parameters in any of the RDA models displayed VIFs less than 10. The adjusted $R^2$ is provided as goodness-fit-stat. All statistical analyses were conducted in R using the core distribution, version 3.3.2 and R-Studio, version 1.0.153,with following packages: vegan [29], lmerTest for the GLMM [24], ALDEx2 [29] and multcomp [28].

# 3. Result

## 3.1 Environmental characterization

Environmental parameters (pH, salinity) and nutrients (including DOC, TN, NOx, nitrate, DIN, phosphate: $PO_4^{3-}$-P, and silicate: $Si(OH)^{4-}$) concentrations for all samples were measured (Table 1). Samples were slightly alkaline (pH 8.0 to 8.7) in nature. The GLMMs analysis indicated that the measured water nutrients that differed significantly among the two sampling station were mainly TN, DIN and $PO_4^{3-}$-P (Table 1; S1 Table). The PCA ordination (Fig 2) showed that first two principal components (PC1 and PC2) represented 74.4% of data variation among sites. PC1 alone represents 60.1% of total variation and influenced by most of the measured parameters, while pH showed a strong correlation with PC2 (S1 Fig). Noteworthy to mention, the samples were separated into two clusters by PC1. One cluster is mainly composed with the samples from ISM (except TH2.3 of OMW) and other cluster accommodating samples collected from OMW. This ordination probably indicates elevated eutrophication in samples from the OMW compared to ISM.

## 3.2. Microbial communities

Total numbers of reads generated per sample ranged between 50517 to 90468 (after merged) corresponding to 3,644 to 6,470 swarmed, non-singleton 16S OTUs (S2 Fig). After rarefaction, numbers of bacterial and archaeal OTUs ranged between 3390 to 5415 and 37 to 91, respectively (S2 Fig). None of the measured diversity indices (Average Shannon diversity and inverse Simpson indices) were found significance differences in between OMW and ISM (Fig 3; S2 Table) indicated by Kruskal—Wilcoxon test ($p > 0.5$), although values varied considerably.

The microbial community of marine estuary water from Sundarbans was dominated with bacteria occupying more than 96% of total community and archaea represented only 4%. Bacterial assemblage of two different sites this area showed to have a distinct community. In class

**Table 1. Environmental conditions in Indian Sundarban mangroves and details of the sampling sites.**

| Station id | Site Id | GPS (DD COORDINATES) | Sample ID | Salinity | pH | DOC (µM) | TN (µM) | NOx (µM) | NO₃ (µM) | DIN (µM) | PO₄ (µM) | Si (µM) |
|---|---|---|---|---|---|---|---|---|---|---|---|---|
| Open Marine Water (OMW) | TH 2 | N 21.82389 | TH2.5 | 26.6 | 8.2 | 112.6 | 12.5 | 1 | 1 | 1 | 0.4 | 11.7 |
| | | E 88.50585 | TH2.2 | 27.0 | 8.2 | 111 | 11.4 | 1.4 | 1.4 | 1.5 | 0.2 | 13.1 |
| | | | TH2.3 | 25.7 | 8.3 | 66.2 | 6.4 | 0 | 0 | 0 | 0.1 | 7.7 |
| | KL 2 | N 21.8172 | KL2.3 | 26.6 | 8.3 | 129.5 | 13.6 | 2.6 | 2.5 | 2.6 | 0.2 | 14.3 |
| | | E 88.53658 | KL2.6 | 26.9 | 8.1 | 162.4 | 10 | 1.2 | 1.2 | 1.3 | 0.1 | 26.3 |
| | | | KL2.1 | 26.3 | 8.2 | 116.3 | 12.1 | 1.5 | 1.4 | 1.5 | 0.6 | 15.9 |
| | BL 1 | N 21.78962 | BL1.6 | 26.8 | 8.5 | 121.1 | 14.8 | 1 | 1 | 1.1 | 0.2 | 16.3 |
| | | E 88.50534 | BL1.5 | 26.9 | 8.4 | 114.1 | 13.2 | 1.7 | 1.7 | 1.8 | 0.2 | 12.6 |
| | | | BL1.2 | 26.5 | 8.7 | 111.4 | 12.7 | 2.3 | 2.2 | 2.3 | 0.2 | 16.1 |
| Island of Sundarban Mangroves (ISM) | KL 1 | N 21.85189 | KL1.3 | 23.6 | 8.3 | 56.1 | 4.4 | 0.2 | 0.2 | 0.3 | 0.1 | 6.2 |
| | | E 88.51168 | KL1.6 | 22.9 | 8.0 | 58 | 6 | 0.9 | 0.8 | 0.9 | 0.1 | 6.1 |
| | | | KL1.2 | 24.3 | 8.1 | 62 | 5.1 | 0.2 | 0.2 | 0.2 | 0.1 | 6.5 |
| | KP 1 | N 21.85604 | KP1.6 | 25.8 | 8.1 | 78.5 | 5.7 | 0 | 0 | 0.1 | 0.1 | 12.2 |
| | | E 88.51191 | KP1.1 | 25.3 | 8.1 | 92.5 | 9 | 0 | 0 | 0.1 | 0.2 | 15.4 |
| | | | KP1.3 | 24.9 | 8.1 | 81.2 | 7 | 0.4 | 0.4 | 0.5 | 0.1 | 12.3 |
| | TH1 | N 21.85706 | TH1.4 | 21.5 | 8.3 | 91.9 | 8.3 | 0 | 0 | 0.1 | 0.2 | 8.7 |
| | | E 88.51638 | TH1.3 | 21.8 | 8.3 | 86.8 | 8.9 | 0.1 | 0 | 0.1 | 0.1 | 8.7 |
| | | | TH1.2 | 21.2 | 8.3 | 84.8 | 7.4 | 0.6 | 0.6 | 0.6 | 0.1 | 9.5 |

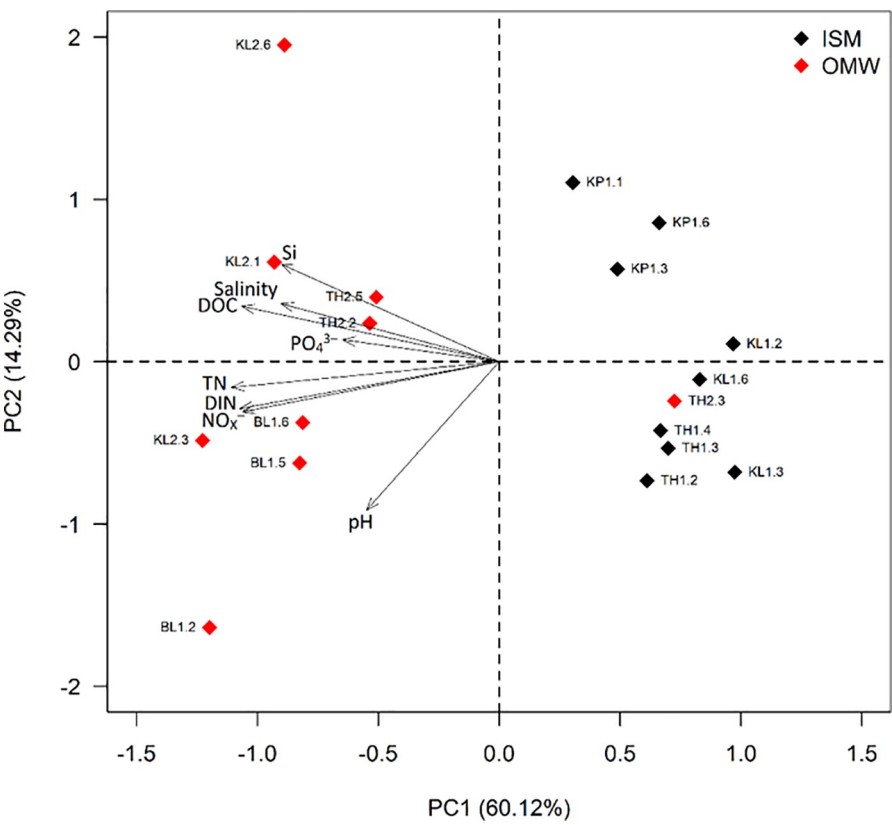

**Fig 2. Principal component analysis (PCA) to ordinate the eighteen collected water samples collected samples from ISM and OMW based on their environmental parameters.** The arrows show the direction of the environmental parameters. DIN, dissolved inorganic nitrogen; TN, total nitrogen; DOC, dissolved organic carbon.

level, among the dominant bacterial groups, *Flavobacteria* (ISM: 15.9% vs OMW: 8.6%), *Alphaproteobacteria* (ISM: 29.5% vs OMW: 28%), and *Acidimicrobiia* (ISM: 6.6% vs OMW: 5.0%) were dominant in ISM while OMW was dominated with mainly with *Gammaproteobacteria* (ISM: 22.6% vs OMW: 35.3%) (**S3 Fig**). At higher taxonomic resolution levels (**Fig 4**),

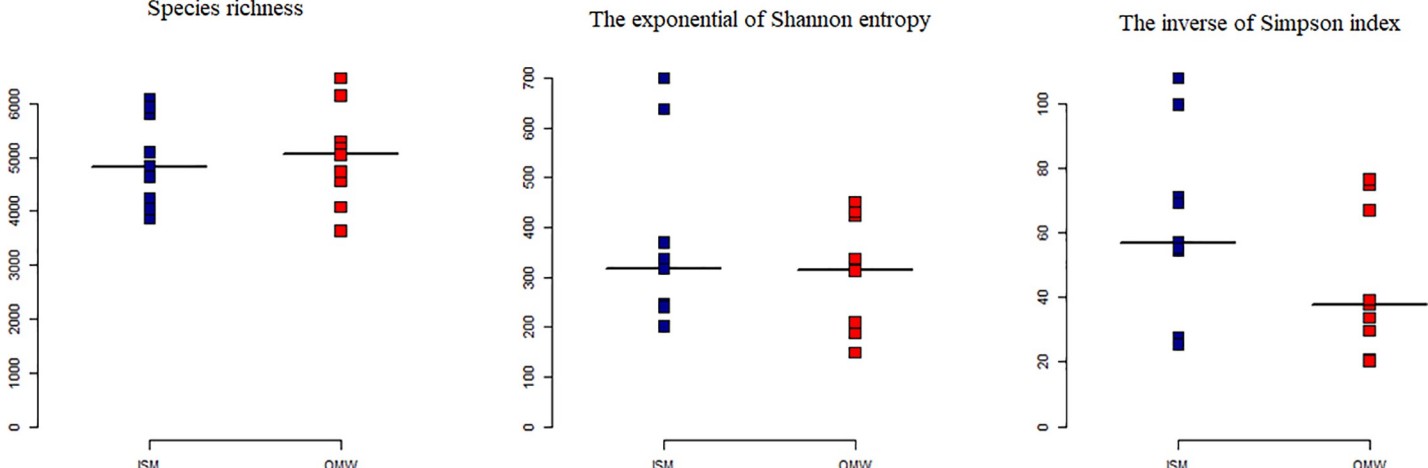

**Fig 3. Alpha diversity of the water microbial community at two different sites (ISM and OMW) of Sundarban mangrove forest areas.** Values are calculated based on repeated random subsampling to the lowest number of sequences per sample. The median per group presented in black line.

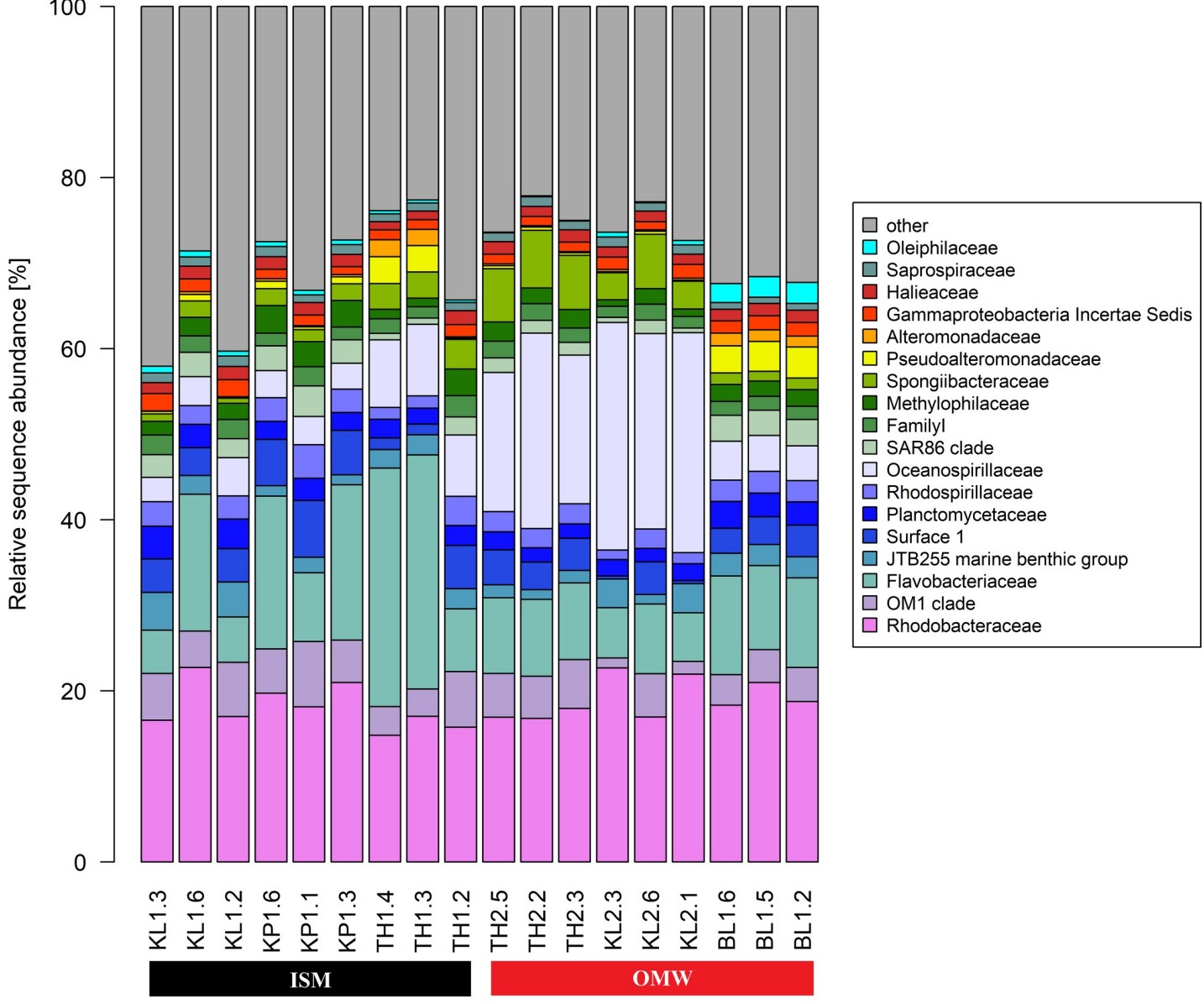

**Fig 4. Taxonomic composition of dominant bacterial taxa on family level across eighteen samples under sites ISM and OWM (nine samples each).** Ten (10) most abundant bacterial families for each of the samples were reported here and rests less dominant members are label as "other".

bacterial communities were composed with a total of 474 and 915 different bacterial family and genus, respectively. The most dominant bacterial family was *Rhodobacteraceae* (18.6%), almost equally distributed between studied two sites. Other dominant bacterial families of ISM were *Flavobacteriaceae* (14.8%) and OM1 clade (5.2%) whereas in OMW, *Oceanospirillaceae* (16%) and *Spongiibacteraceae* (4%) were the most abundant.

We observed dominancy of Marine Group (MG I) (currently known as *Thaumarchaeota*) and *Euryarchaea* MG II in archaeal community assemblages with 78.5% and 16.9% of relative abundance, respectively while the presence of *Woesearchaeota* (2.1%) was also evident **(Fig 5)**. MG I was found in relatively higher abundant at ISM constituting on average 82.1% of sequences as opposed to 75% at OMW. *Euryarchaeota* MG II comprised about 20.3% at OMW

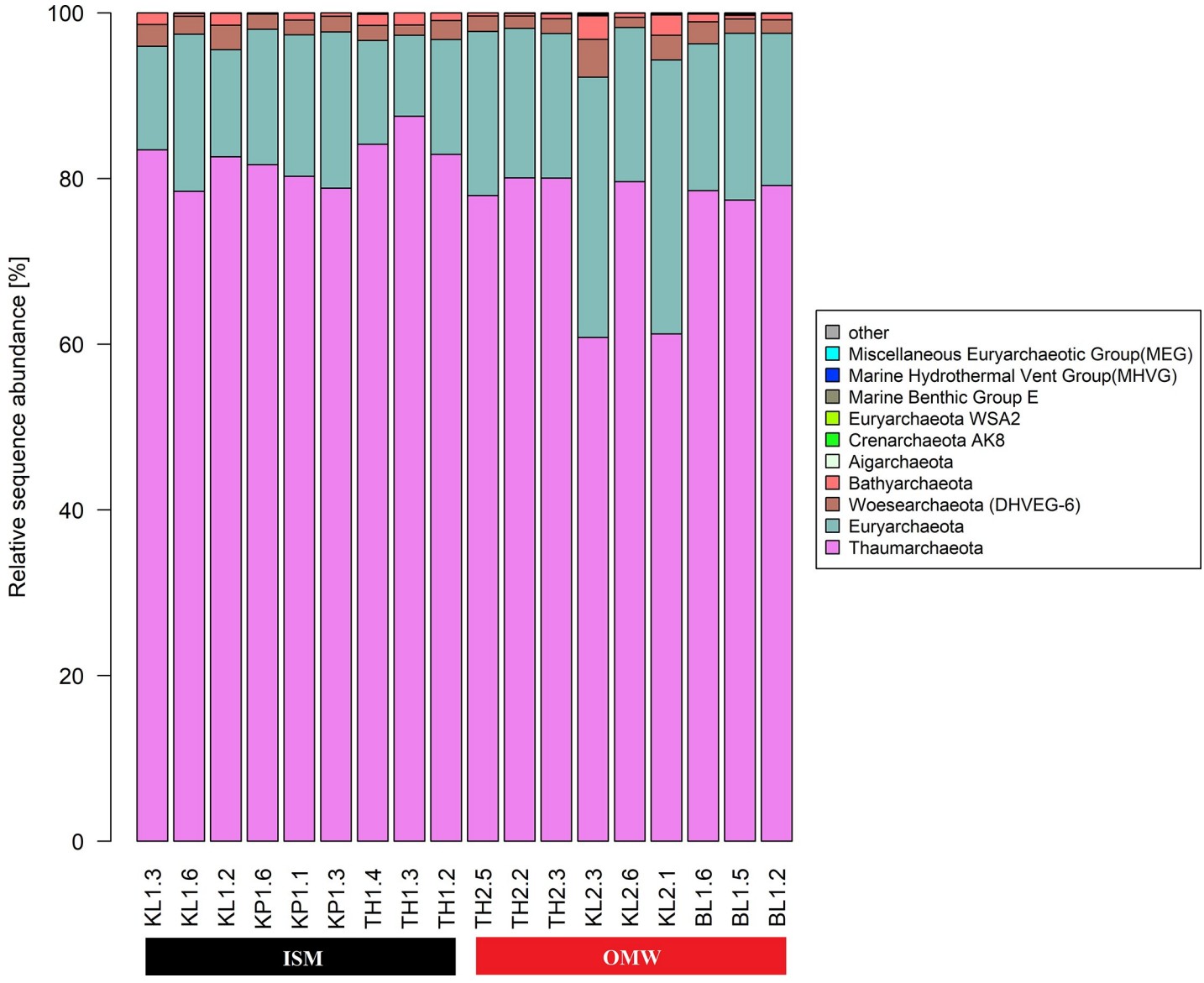

**Fig 5. Taxonomic compositions of dominate archaeal phyla across eighteen samples represents two sites ISM and OWM (nine samples each).** Ten (10) most abundant phyla for each of the samples were reported here and rests less dominant members are label as "other".

compared to 13.7% at ISM. Among total twenty-nine (29) archaeal genera, *Candidatus Nitrosopumilus* and *Candidatus Nitrosopelagicus* accounted for the 40.7% and 21.4% of total relative abundance, respectively (S4 Fig).

### 3.3 Environmental drivers of bacterial communities

At OTU resolution level also, distinct microbial communities were observed between OMW and ISM based on changes in community structure (beta diversity) which is quantified by non-metric multidimensional scaling (NMDS) plot by calculating Bray–Curtis dissimilarly (Fig 6). This pattern is confirmed by the ANOSIM test that indicated a significant difference in microbial community structure between ISM and OMW (ANOSIM, R = 0.24, p < 0.001). Redundancy analyses attempted to identify the water quality parameters that had strong

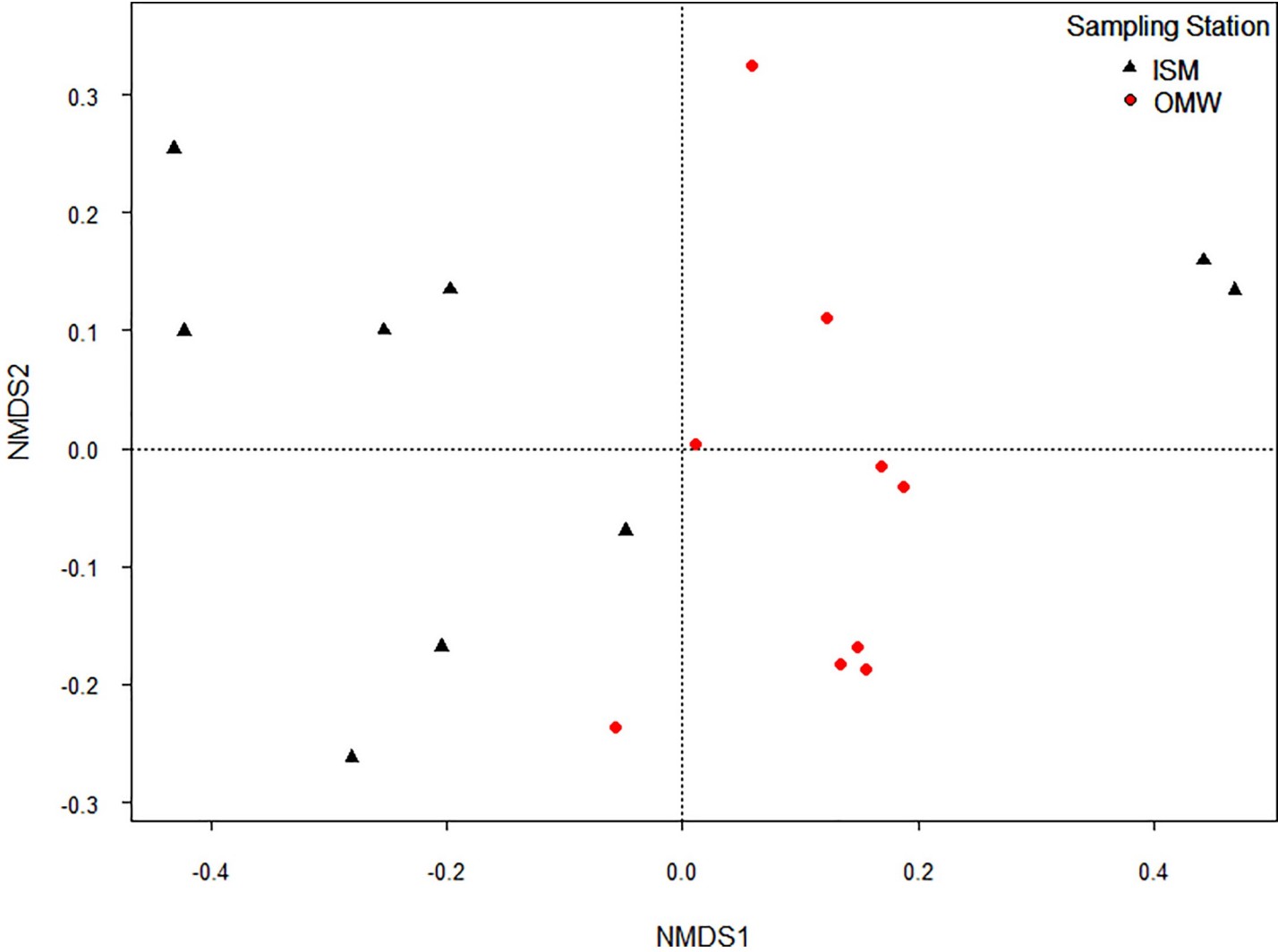

**Fig 6. Non-metric multidimensional scaling (NMDS) plot of bacterial community composition of the bacterial communities of each sampled at the inhabited island (ISM) and open marine areas (OWM).**

explanatory power for microbial communities. We observed that total nitrogen (TN) and dissolved inorganic nutrients (DIN) accounted for almost 10% of the variability in microbial community of OMW where TN alone explain 6% variation of microbial community (RDA, $R^2 = 0.06$, $F_{(1,7)} = 1.34$, $p < 0.05$). In contrast, salinity and TN explained approximately 9% of the variability in community composition of ISM (RDA, $R^2 = 0.06$, $F_{(1,7)} = 1.34$, $p < 0.05$) and alone salinity responsible for explaining 7% microbial variation of this site (**Table 2**).

## 4. Discussion

The pH values (8.0–8.7) indicates the water of ISM and OMW slightly alkaline in nature which supports the previous findings in similar samples from Sundarbans Mangrove forest areas [30]. Such ranges of pH may be attributed by the buffering capacity of water that support high biological activity [31,30]. The water of Sundarbans is characterized by elevated salinity values in line with previous reports [31]. The long-term changes in water properties in the eastern part of Sundarbans, sampling regions of our study, indicating increased trends on salinity and

**Table 2. Contribution of environmental parameters including nutrient content of six sampling sites to explaining the variation in microbial community composition based on redundancy analysis (RDA).**

| Sampling station | Explanatory variable | Adjusted $R^2$ | F | df | p-Value |
|---|---|---|---|---|---|
| OMW | TN + DIN | 6% | 1.24 | 2, 6 | 0.022* |
| | TN | 4% | 1.34 | 1, 7 | 0.029* |
| | DIN | - | 1.40 | 1, 7 | 0.125 |
| ISM | Salinity + TN | 7% | 1.31 | 2,6 | 0.043* |
| | Salinity | 9% | 1.77 | 1,7 | 0.004** |
| | TN | 0.6% | 1.05 | 1,7 | 0.326 |

p-Values defined as significant at a threshold of 0.05 are highlighted in asterisks mark

Adjusted $R^2$ are provided as goodness-of-fit metrics

df degrees of freedom (numerator, denominator).

pH [3]. Our result shows differences in measured environmental parameters between two sites represented by three sampling stations and nine samples each leading to their segregation into two clusters (in PCA analysis) along with their sampling sites. This ordination as a result of different nutrient loads and this is reflected by potential eutrophication in water from OWM. Influences from the Thakuran and Matla rivers reported to have a strong impact on the estuary water quality represented by OWM [4,5], that also reflected in our study. The perennial discharge in Thakuran and Matla Estuary from upstream regions brings in a high suspended matter load throughout the year [30]. Those estuaries severely contaminated with huge organic load and sediment flux originated from upstream domestic sewage, aquaculture, intensive trawling activities, agricultural runoff as well as soil erosion [32,33] may supports the prospective cause for eutrophication in the OWM sites.

Because of the relevance of microbial community of Indian Sundarbans, several investigators attempted on surface sediments samples [34,8,3,10,35] as well as recently on water column of this regions [12,13] using 16S rRNA gene metagenomic approaches. Unlike previous studies, this investigation attempted to asses both the bacterial and archaeal community at a same time of water from relative less anthropogenic disturbance sites using an efficient primer set to target V4-V5 variable region of 16S rRNA gene in order to avoid the problems of underestimated or overestimated common marine taxa [14], therefore our investigation gives more accurate and well-resolved picture of microbial communities of these sites.

Although insignificant differences, elevated trends of α-diversity of the marine estuary water samples (OWM) might be an indications of relatively rich bacterial community compared to ISM of Sundarbans might be attributed toward their elevated eutrophication. This observation was supported by previous reports that indicate a higher diversity and equitability in the human impacted estuary because of proliferation of several different microorganisms [36,37].

The bacterial assemblages of the studied samples (specially OMW) showed similarity with the previously reported bacterial community of marine sediments and water samples of Sundarban Mangrove areas [8, 10–12]. The most dominant bacterial family presents both the station with almost equal proportion is *Rhodobacteraceae*. Dominancy of members of this family in marine water microbial community previously reported and known to be associated with marine phytoplankton blooms where it plays important role in transforming phytoplankton-derived organic matter [34,38,39]. The abundant OTUs of *Rhodobacteraceae* are classified as anoxygenic phototrophs *Nautella*, reported to serve an indicator of marine eutrophication, predominantly found in higher eutrophic OMW samples. The other dominant one under the

same family is marine heterotrophs *Ruegeria*s (almost equally distributed among both ISM and OMW) serve as the model of marine sulfur and carbon cycle [38,40,41].

Interestingly enrichment of several OTUs from the *Flavobacteria* is observed in the oligotrophic ISM compared to the eutrophic OMW. They are specialized in utilization of biopolymers and organic substances in oligotrophic environment i.e., when organic substances present even at very low concentrations. Higher abundance of these polymers degrading bacteria responsible for releasing the nutrients heterotrophs/copiotrophic organisms thus helps in the growth of heterotrophic bacterial community at oligotrophic marine environments [42–44]. Therefore, they might plays central role for balancing the heterotrophs/copiotrophic microbial ecology of ISM. The dominant OTUs under family *Flavobacteriaceae* were mainly classified as *Aureimarina* and NS5 marine group genus. Roles of *Aureimarina* in marine biogeochemistry has not been investigated much although few studies reported on their presences in coastal seawater and saline estuarine [45,46]. However, this is the first report of their abundance in marine water of Sundarbans. The NS5 marine group which are equally dominated in both the studied sites are reported to be ubiquitous in the seawater-related samples and known for phytoplankton-derived macromolecules [47–49].

This investigation identified *Actinobacteria* constituted a predominant fraction both in OMW and ISM but elevated amount in the later samples. Bacteria under this group are consisted of both copiotrophic and oligotrophic members with higher abundance in oligotrophic marine environments [50,51]. Their presence in mangrove estuary regions is well documented [52–54]. As marine *Actinobacteria* are the richest sources of secondary metabolites thus, have been well reported as potential sources of bioactive compounds [55]. Therefore, their abundance in our studied sites (specially ISM) would be potential hotspot for isolating bioactive molecules from Indian Sundarban mangrove forest. The OM1 clade (dominating in ISM), an uncultured Actinobacterial clade, frequently recovered from various marine environments with higher abundance at near coastal sites than open marine areas however supports our reports [56–58]. The dominant OTUs of this family were classified as *Candidatus Actinomarina*. Those photoheterotrophs are one of smallest free-living prokaryotes are reported to be ubiquitous in marine systems. Not many reports are described their role in the marine biogeochemical cycle but *Actinobacteria* are reported to be associated in carbon cycling to decompose the plant biomass via degrading the cellulose and hemicellulose materials, a dominate resources materials in mangrove, however may supports their dominancy in ISM. They are also known as chitin, hydrocarbons and organic contaminants degrader [59,60].

The OTUs affiliated to families *Oceanospirillaceae* and *Spongiibacteraceae* of *Gammaproteobacteria* showed increase abundance in the impacted site OMW. Bacteria from these families are known to be present in eutrophic marine environments. They are known as polymer degraders and can utilize polyhydroxy alkanoate compounds and proteorhodopsin, for harvesting an additional energy, supports their living in eutrophic water samples [61–63]. The dominant OTUs of *Oceanospirillaceae* are affiliated to chemoheterotrophic genus *Marinobacterium*. Their presence in mangroves as well as surface seawater have already been described in previous studies and known to be associated with hydrocarbon biodegradation [64,65]. The other dominated bacterial family in the samples from OMW is *Spongiibacteraceae*. They comprise mainly marine bacteria known as Oligotrophic Marine *Gammaproteobacteria* (OMG) group [66,67]. We recorded the dominant OTUs of this family are affiliated with BD1-7 which is a cosmopolitan group of *Gammaproteobacteria* is mostly autochthonous, reported to inhabits at diverse marine habitats [68,65,69]. In line with previous reports this investigation, therefore, indicates proliferation of bacterial groups under *Gammaproteobacteria* with respond to increased nutrient concentrations in estuary [36].

However, in contrast to sediments reported in previous investigations, an archaeal community of marine waters in the Sundarban mangroves is dominated with *Thaumarchaeota* Marine Group (MG I) and *Euryarchaea* MG II. The chemolithoautotrophic MG I which are in higher in number on ISM are responsible for oxidation of ammonia and showed ability in inorganic carbon fixation [70] thus important players in global Carbon (C) and Nitrogen (N) biogeochemical cycles. While the heterotrophic MG II, dominated in OMW, previously reported to be abundant in the marine aquatic environment [71]. Their abilities to acquired energy in presences of light through organic carbon degradation in the photic zone is also documented [71]. Dominant OTUs of MGI group are affiliated with ammonia-oxidizing archaeal, Candidatus Nitrosopumilus and Candidatus Nitrosopelagicus, play important roles in nitrogen and carbon cycling of marine ecosystem [3]. However, this investigation reports first on their presence in this areas. Therefore, the biological and geochemical processes at estuary water habitats in the Indian Sundarban Mangrove areas have likely influenced the archaeal community structure.

Overall, our study indicates along with the elevated level of average pH and salinity, the open marine water (OMW) showed eutrophication probably leads to an observed bacterial shift toward more copiotrophic and photoheterotrophic bacterial (Oceanospirillaceae and Spongiibacteraceae) and archaeal community (Euryarchaea MG II) and compared to the more oligotrophic microbial community (Aureimarina, NS5 marine group, OM1 clade and Thaumarchaeota MG I) of costal water of a small Island of Sundarban Mangroves (ISM). These microbial assembles thus might represent key players in biogeochemical cycle of this mangrove and the studied areas represent a hotspot for bacterial having potential to produce the commercially important secondary metabolites. This investigation also reports that total nitrogen and dissolved inorganic nitrogen are the major environmental contributors on determining the microbial communities for OMW and for ISM it is combination of total nitrogen and salinity.

## 5. Conclusion

This investigation provides the first details description of bacterial and archeal communities concurrently of Thakuran and Matla river complex of Maipith coastal areas in the Indian Sundarbans mangroves areas. This study indicates the eutrophication in open marine water (OMW) dominated with more copiotrophic and photoheterotrophic bacterial and archaeal community (Euryarchaea MG II) and while oligotrophic microbial community abundant in costal water of the Island of Sundarban Mangroves (ISM).

However, given the rising burden on Indian mangrove coastal ecosystems, this study suggests that sewages from urban areas lacking proper treatment can alter microbial communities that may play vital role in biogeochemical cycle (nitrogen cycle) of mangrove ecosystem and consequently may impact on the climate in the tropical country.

## 6. Acknowledgments

Foremost PKD would like to thank NAM S&T Centre New Delhi, India and ZMT, Bremen for selection. PKD also acknowledge to Jadavpur University for granting the leave to avail this fellowship. We sincerely thank the faculty and technical staff of ZMT Bremen for their generous help to carry this research work. Many thanks to Dr Halina Tegetmeye of CeBiTec Bielefeld, Germany for sequences. The assistance of conducting analyses at the laboratories at the ZMT in Bremen by Matthias Birkicht, Sonja Peters and Achim Meyer also acknowledge. Dr. Debajit Datta, Assistant professor, Jadavpur University is also acknowledged for helping in designing the map of the sampling location.

## Supporting information

**S1 Fig. Heatmap of pairwise correlations between the different environmental parameters and three principal components.** Levels of correlations are indicated with different color bar. (TIFF)

**S2 Fig. The rarefaction curve of the eighteen (18) samples, indicated by the number of OTUs as a function of the number of reads.** The curve approaching plateau indicates that the number of reads are enough to describe the OTUs representing the community. (TIFF)

**S3 Fig. Taxonomic composition of the ten (10) most abundant bacterial phyla in the studied two sites ISM and OWM represented by eighteen samples (nine samples each).** (TIF)

**S4 Fig. Taxonomic composition of the ten (10) most abundant archaeal genus in the studied two sites ISM and OWM represented by eighteen samples (nine samples each).** (TIF)

**S1 Table. Kruskal-Wallis test for Environmental parameters at the six sampling sites.** (DOCX)

**S2 Table. Kruskal-Wallis test for alpha diversity of the six sampling sites.** (DOCX)

## Author Contributions

**Conceptualization:** Paltu Kumar Dhal.

**Data curation:** Paltu Kumar Dhal.

**Formal analysis:** Paltu Kumar Dhal.

**Funding acquisition:** Paltu Kumar Dhal, Astrid Gärdes.

**Investigation:** Paltu Kumar Dhal.

**Project administration:** Paltu Kumar Dhal, Astrid Gärdes.

**Supervision:** Paltu Kumar Dhal, Astrid Gärdes.

**Writing – original draft:** Paltu Kumar Dhal.

**Writing – review & editing:** Germán A. Kopprio, Astrid Gärdes.

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
