## [Decision Letter · Decision Letter 0]

4 Oct 2019

PONE-D-19-22253

Insights on aquatic microbiome of the Indian Sundarbans mangrove areas

PLOS ONE

Dear Dr. Dhal,

Thank you for submitting your manuscript to PLOS ONE. After careful consideration, we feel that it has merit but does not fully meet PLOS ONE’s publication criteria as it currently stands. Therefore, we invite you to submit a revised version of the manuscript that addresses the points raised during the review process.

We would appreciate receiving your revised manuscript by Nov 18 2019 11:59PM. To enhance the reproducibility of your results, we recommend that if applicable you deposit your laboratory protocols in protocols.io, where a protocol can be assigned its own identifier (DOI) such that it can be cited independently in the future. For instructions see: http://journals.plos.org/plosone/s/submission-guidelines#loc-laboratory-protocols

We look forward to receiving your revised manuscript.

Kind regards,

Jiang-Shiou Hwang, Ph.D.

Academic Editor

PLOS ONE

**Journal Requirements:**

2. We note that  Figure(s) 1 in your submission contain [map/satellite] images which may be copyrighted. All PLOS content is published under the Creative Commons Attribution License (CC BY 4.0), which means that the manuscript, images, and Supporting Information files will be freely available online, and any third party is permitted to access, download, copy, distribute, and use these materials in any way, even commercially, with proper attribution. For these reasons, we cannot publish previously copyrighted maps or satellite images created using proprietary data, such as Google software (Google Maps, Street View, and Earth). For more information, see our copyright guidelines: http://journals.plos.org/plosone/s/licenses-and-copyright.

a) You may seek permission from the original copyright holder of Figure(s) [#] to publish the content specifically under the CC BY 4.0 license.  

**Comments to the Author**

1. Is the manuscript technically sound, and do the data support the conclusions?

Reviewer #1: Partly

Reviewer #2: Partly

2. Has the statistical analysis been performed appropriately and rigorously? 

Reviewer #1: No

Reviewer #2: Yes

3. Have the authors made all data underlying the findings in their manuscript fully available?

Reviewer #1: No

Reviewer #2: Yes

4. Is the manuscript presented in an intelligible fashion and written in standard English?

Reviewer #1: Yes

Reviewer #2: Yes

5. Review Comments to the Author

Reviewer #1: PLOS ONE

Manuscript: PONE-D-19-22253

In this manuscript, the authors examined environmental parameters, microbial community composition in the island of Sundarban mangroves (ISM) and open marine water (OMW) using 16S rRNA gene-based amplicon, and attempted to assess the capability of environmental parameters for explaining the variation in microbial community composition. The authors reported that a higher eutrophic condition changes in bacterial communities in OMW. The authors did not make a solid conclusion. The authors concluded that multiple environmental/anthropogenic stressors (salinity, pollution, eutrophication, land-use) affect the estuary water and consequently the microbial communities in concert. Suggest also sample open seawater (or seawater standard), conduct examinations of water parameter and microbial community in open seawater, ISM, and OMW, and make comparison among three areas, and make full discussion. Suggest report clearly on pollution and eutrophication (Line 54). Examination on the environmental determinants that contribute to the microbial communities is unclear (Line 103-104). Suggest examine total phosphorus, organic phosphorus, total alkalinity, hardness, dissolved solids, and major ions and substances like sodium, potassium, calcium, magnesium, sulfate, carbonate, and chloride. There are many grammatical errors and typo errors.

1. Line 53: Rewrite the sentence.

2. Line 112, Line 117: Reference “Das 2011” is not listed in the Reference section.

3. Lines 117-119: Suggest examine total phosphorus (TP). Suggest examine orthophosphate-phosphorus (Orthophosphate-P, PO43--P).

4. Lines 127-136: Suggest rewrite the sentences. Suggest write detail about each parameter assay. It said “continuous flow analyzer (Flowsys by Unity Scientific, Brookfiels, USA)” (Line 129-130). It said “spectro-photometrical analysis with a Flows continuous flow analyzer (Systea, Anagi, Italy)” (Lines 133-134).

5. Line 129: It is “phosphate (PO43-)” or “orthophosphate (PO43-).

6. Line 128: Report content of dissolved inorganic nutrients (DIN).

7. Line 128: Report clearly on nitrate and nitrite (NOx).

8. Line 129: The statement “silicate (Si)” is not correct. Report it is silicon (Si), Silica (SiO2), or silicate.

9. Line 132: Suggest report examination of major water compositions like sodium, potassium, calcium, magnesium, sulfate, bicarbonate, carbonate, chloride.

10. Line 156: Reference “Maher et al 2014” is not listed in the Reference section.

11. Line 265: Suggest report clearly on high suspended matter.

12. Line 272: Reference “Eloc-Fadrosh et al. 2015” is not listed in the Reference section.

13. Line 273: Change to “Ghosh and Bhadury 2018, 2019”.

14. Lines 288-289: Change to “Basak et al. 2015, 2016; Ghosh and Bhadury 2017, 2018”.

15. Line 203: Reference “Buchan et al. 2010” is not listed in the Reference section.

16. Lines 302-306: Rewrite the sentence.

17. Lines 351-356: Rewrite the sentences.

18. Lines 251-358: The discussion is disjointed. Suggest link water parameters and microbial community composition in Family, make full discussion, and make discussion in these areas and open seawater.

19. Lines 359-377: The conclusion is lengthy. Suggest make a solid conclusion.

20. Lines 394-540: Suggest follow the Journal format, and write reference. Suggest write all authors.

Reviewer #2: In this study the authors have investigated aquatic microbial community structure from two estuaries of Sundarbans, world's largest contiguous mangrove ecosystem. While the findings of this study are potentially very interesting, I have highlighted some points that the authors need to address along with further strengthening of the discussion section.

The reference cited in the first line of Introduction is not correct. For example, Spalding et al (1997) and Gopal and Chauhan (2006) references would be more appropriate.

Introduction: However, sampling locations of those studies on microbial communities.... a single site specific with limited number of samples. This sentence need to be correct. The paper by Ghosh and Bhadury (2019) has looked in bacterioplankton community composition from four estuaries- Mooriganga, Thakuran, Matla and Harinbhanga.

Was the sampling undertaken during high or low tide? This should be clearly mentioned. What factors were taken into consideration for selection of sites? These points should be clearly stated.

How was the calibration of Eureka 2 Manta multiprobe undertaken?

Why V4-V5 region of 16S rRNA was targeted? This needs to be written upfront in introduction section given the importance of scales of taxonomic resolution among variable regions of this molecule.

The authors have reported dominance of Actinobacteria-like OTUs/sequences in their studied samples. I believe it is important to compare this finding with previous study undertaken in the same estuary as well as from other mangroves (e.g. Gong et al 2019, Scientific Reports). It is also important to highlight the specific roles of this group in terms of biogeochemical cycling (e.g. breakdown of complex forms of carbon and resulting role in remineralization).

One of the important groups that was reported from this mangrove is the presence and dominance of Firmicutes. However, I do not see any discussion on this point given Firmicutes signal were found to be much higher in Sundarbans compared to other mangroves.

It is really interesting to see the relative dominance of Thaumarchaeota Marine Group (MG I) and Euryarchaea MG II in this ecosystem. I am just curious if the authors can link this finding with reported dissolved ammonium concentration in this sector of Sundarbans given the availability of ammonia seems to be episodic in nature and thus there are consequences for nitrification rates. Also, the selection of primers improved the resolution in terms of encountering these groups? It is also important to highlight given the sampling has been focused only one time point, the observed community structure may also exhibit temporal variability.

6. PLOS authors have the option to publish the peer review history of their article (what does this mean?). If published, this will include your full peer review and any attached files.

Reviewer #1: No

Reviewer #2: No

---

## [Author Response · Author response to Decision Letter 0]

20 Dec 2019

Journal Requirements

Q 1. When submitting your revision, we need you to address these additional requirements.

http://www.journals.plos.org/plosone/s/file?id=wjVg/PLOSOne_formatting_sample_main_body.pdf. and http://www.journals.plos.org/plosone/s/file?id=ba62/PLOSOne_formatting_sample_title_authors_affiliations.pdf.

Answer: We have tried to make all necessary changes as needed.

Q 2. We note that Figure(s) 1 in your submission contain [map/satellite] images which may be copyrighted. All PLOS content is published under the Creative Commons Attribution License (CC BY 4.0), which means that the manuscript, images, and Supporting Information files will be freely available online, and any third party is permitted to access, download, copy, distribute, and use these materials in any way, even commercially, with proper attribution. For these reasons, we cannot publish previously copyrighted maps or satellite images created using proprietary data, such as Google software (Google Maps, Street View, and Earth). For more information, see our copyright guidelines: http://journals.plos.org/plosone/s/licenses-and-copyright.

Answer: Thanks for rising this issue. You may notice the above said Figure (i.e Figure 1) is a map of the sampling sites. This is to bring your kind notice that this figure neither copied / collected from any other sources nor it is a previously copyrighted maps. The concern figure has been prepared manually in laboratory using open source database of GADM (https://gadm.org/). The GPS data of sampling sites were collecting during sampling and those were compiled together in an ArcGIS10.3 software environments. Therefore, I think this figure do not need any copywriter permission. The revised manuscript is updated with this information in legend section of the Figure 1. 

5. Review Comments to the Author

Reviewer #1: 

P1. In this manuscript, the authors examined environmental parameters, microbial community composition in the island of Sundarban mangroves (ISM) and open marine water (OMW) using 16S rRNA gene-based amplicon, and attempted to assess the capability of environmental parameters for explaining the variation in microbial community composition. The authors reported that a higher eutrophic condition changes in bacterial communities in OMW. The authors did not make a solid conclusion. The authors concluded that multiple environmental/anthropogenic stressors (salinity, pollution, eutrophication, land-use) affect the estuary water and consequently the microbial communities in concert.

Answer: We agree with the reviewer comments that this manuscript may need more concrete conclusion. The microbial communities of world largest mangrove sites demands more extensive investigations. However, our investigation few first attempt to reports the changes of microbial community that majorly impacted by the multiple environmental/anthropogenic stressors of the mangrove estuary water. In line with the reviewer suggestions we have included few more important observations on microbial communities in the discussion part of current manuscript. (Line 262-262, Line 295-299, Line 310-325, Line 348-354, Line 357-367)

P2. Suggest also sample open seawater (or seawater standard), conduct examinations of water parameter and microbial community in open seawater, ISM, and OMW, and make comparison among three areas, and make full discussion.

Answer: Thanks for this rising this issue and we also agree with the reviewer suggestions to on include the samples from open seawater. In this connection, this is to bring your notice that, OMW represents the samples collected the open marine water were included in the current investigation. The water parameters and microbial community of OMW were measured and compared with ISM which is marine water collected from coastal regions of a small Island of Sundarban Mangroves.

P3. Suggest report clearly on pollution and eutrophication (Line 54).

Answer: Thanks for suggestions. Now in the current manuscript updated information on the pollution and eutrophication is clearly stated (Lines 262)

P4. Examination on the environmental determinants that contribute to the microbial communities is unclear (Line 103-104).

Answer: In microbial ecology, redundancy analysis (RDA), a multivariate statistical analysis, is considerably being used to determine which of the environmental factors are the most significant to explain variation in microbial community composition (Alban Ramette, 2007; Chen et al 2018; Lee et al 2019; Shankar et al 2019). However, in the present study we have also attempted to evaluate the role of environmental parameters as drivers of the variations in community compositions in the studied sites. Our result indicates, total nitrogen (TN) and dissolved inorganic nitrogen (DIN) accounted for almost 10% of the variability in microbial community of Open Marine Water where TN alone explain 6% variation of microbial community (RDA, R2 = 0.06, F(1,7) = 1.34, p < 0.05). In contrast, salinity and TN explained approximately 9% of the variability in community composition of the water collected from costal area of an small Island of Sundarban Mangroves (RDA, R2 = 0.06, F(1,7) = 1.34, p < 0.05) and alone salinity responsible for explaining 7% microbial variation of this site .

P5. Suggest examine total phosphorus, organic phosphorus, total alkalinity, hardness, dissolved solids, and major ions and substances like sodium, potassium, calcium, magnesium, sulfate, carbonate, and chloride.

Answer: We would also like to thanks the reviewers for rising this issue. Those are the water parameters that should be measured. But unfortunately in the current situation we can’t do so. All those works had been perform at The Leibniz-Zentrum für Marine Tropenforschung in Bremen, Germany (ZMT) under expert supervisions. After completion of all the experiments were over we were asked to discard all remaining water samples, as per The National Biodiversity Authority (NBA), Govt. of India rules. Therefore, we do not have any more samples to measure the suggested water parameters. 

Q1. Line 53: Rewrite the sentence.

Answer: The mentioned sentence rewritten now.

Q2. Line 112, Line 117: Reference “Das 2011” is not listed in the Reference section

Answer: We agree with the reviewer views and sorry for this typo. It would be “Das 2016” and necessary changes has been made.

Q3. Lines 117-119: Suggest examine total phosphorus (TP). Suggest examine orthophosphate-phosphorus (Orthophosphate-P, PO43--P).

Answer: Thanks for your kind suggestions, but I afraid I can measure total phosphorus (TP) and orthophosphate-phosphorus (Orthophosphate-P, PO43—P) in the current perspective as mentioned previously (lacking of the samples). 

Q4. Lines 127-136: Suggest rewrite the sentences. Suggest write detail about each parameter assay. It said “continuous flow analyzer (Flowsys by Unity Scientific, Brookfiels, USA)” (Line 129-130). It said “spectro-photometrical analysis with a Flows continuous flow analyzer (Systea, Anagi, Italy)” (Lines 133-134).

Answer: Sorry for this unintentional mistake. The lines are corrected now. I was a mistake.

Q5. Line 129: It is “phosphate (PO43-)” or “orthophosphate (PO43-).

Answer: It is phosphate PO43-. We were not dealing with orthophosphate.

Q6. Line 128: Report content of dissolved inorganic nutrients (DIN).

Answer: The dissolved inorganic nutrients (DIN) content already reported in the present manuscript. Their amount varies among the samples as indicated in Table 1 and Figure 2 (PCA plot).

Q7. Line 128: Report clearly on nitrate and nitrite (NOx).

Answer: This manuscript in line with the previous reports (Kegler et al 2018) that uses the NOx to represent the combined amount of nitrate and nitrite content. The nitrite (NO2-) content in the studied samples is very insignificant therefore nitrate (NO3-) content along with NOx were reported in this manuscript (Table 1 and Figure 2).

Q8. Line 129: The statement “silicate (Si)” is not correct. Report it is silicon (Si), Silica (SiO2), or silicate.

Answer: It’s a mistake. Necessary changes has been made as silicate (SiO44−).

Q9. Line 132: Suggest report examination of major water compositions like sodium, potassium, calcium, magnesium, sulfate, bicarbonate, carbonate, chloride.

Answer: I fully agreed with the reviewers suggestions. When the experiments were carried out, we had no plans to measure them as they were not included in the proposed project. Unfortunately in the current situation we cannot measure those parameters because of lacking of samples. 

Q10. Line 156: Reference “Maher et al 2014” is not listed in the Reference section.

Answer: The above mentions citation is “Mahe et al. 2014” which is available in Reference section.

Q11. Line 265: Suggest report clearly on high suspended matter.

Answer: Yes. We fully agreed with the reviewers suggestions. I am again sorry to inform you we are unable to measure any other environmental parameters. 

Q12. Line 272: Reference “Eloc-Fadrosh et al. 2015” is not listed in the Reference section.

Answer: It’s a mistake. The concern citation deleted from the updated manuscript.

Q13. Line 273: Change to “Ghosh and Bhadury 2018, 2019”.

Answer: thanks for this corrections. Necessary change has been made as per suggestion. 

Q14. Lines 288-289: Change to “Basak et al. 2015, 2016; Ghosh and Bhadury 2017, 2018”.

Answer: Necessary correction made in manuscript as suggested. Thank you.

Q15. Line 203: Reference “Buchan et al. 2010” is not listed in the Reference section.

Answer: It’s a typos. It’s would be Buchan et al. 2005.

Q16. Lines 302-306: Rewrite the sentence.

Answer: The sentence are rewritten now.

Q17. Lines 351-356: Rewrite the sentences.

Answer: The sentences in these lines are rewritten now.

Q18. Lines 251-358: The discussion is disjointed. Suggest link water parameters and microbial community composition in Family, make full discussion, and make discussion in these areas and open seawater.

Answer: Thanks for the suggestion. The discussion part of the revised manuscript is updated with modified suggestions. I think, after incorporating the reviewer suggestions, the revised manuscript is now much improved one. (Line 262-262, Line 295-299, Line 310-325, Line 348-354, Line 357-367)

Q19. Lines 359-377: The conclusion is lengthy. Suggest make a solid conclusion.

Answer: Again many thanks for the suggestions. The conclusion part is modified now. 

Q20. Lines 394-540: Suggest follow the Journal format, and write reference. Suggest write all authors.

Answer: We have tried to make all necessary changes as needed

Reviewer #2: 

P1: In this study the authors have investigated aquatic microbial community structure from two estuaries of Sundarbans, world's largest contiguous mangrove ecosystem. While the findings of this study are potentially very interesting, I have highlighted some points that the authors need to address along with further strengthening of the discussion section.

Answer: We fully agreed with the reviewers’ comments and suggestions. We tried to address all the issues raised by the reviewer and this eventually improved the quality of the manuscript.

Q1: The reference cited in the first line of Introduction is not correct. For example, Spalding et al (1997) and Gopal and Chauhan (2006) references would be more appropriate.

Answer: We agreed with the reviewer’s comments and changes has been made in the modified manuscript with the more appropriate citation accordingly.

Q2: Introduction: However, sampling locations of those studies on microbial communities.... a single site specific with limited number of samples. This sentence need to be correct. The paper by Ghosh and Bhadury (2019) has looked in bacterioplankton community composition from four estuaries- Mooriganga, Thakuran, Matla and Harinbhanga.

Answer: Sorry for this mistake and necessary changes have been made in the revised manuscript. (Line 85-89)

Q3: Was the sampling undertaken during high or low tide? This should be clearly mentioned. What factors were taken into consideration for selection of sites? These points should be clearly stated.

Answer: The samples were collected during low tide condition and this information mentioned in the revised MS. The rational for selecting the sites as also stated.

Q4: How was the calibration of Eureka 2 Manta multiprobe undertaken?

Answer: This experiment was performed at ZMT under the expert supervision. This multiprobe kit has the respective calibration solution and calibration was done with those solution. 

Q5: Why V4-V5 region of 16S rRNA was targeted? This needs to be written upfront in introduction section given the importance of scales of taxonomic resolution among variable regions of this molecule.

Answer: Thanks for the suggestions. We agree with reviewer’s view to incorporate justification behind aiming the V4-V5 region of 16S rRNA gene in the introduction section of this manuscript. In line with the reviewers suggestion, the importance of using the primer set (that targeting V4-V5 region of 16S rRNA gene) used in the current investigation has been discussed in the manuscript. (Line 97-101 and Line 267-273)

Q6: The authors have reported dominance of Actinobacteria-like OTUs/sequences in their studied samples. I believe it is important to compare this finding with previous study undertaken in the same estuary as well as from other mangroves (e.g. Gong et al 2019, Scientific Reports). It is also important to highlight the specific roles of this group in terms of biogeochemical cycling (e.g. breakdown of complex forms of carbon and resulting role in remineralization).

Answer: We agreed with the reviewers comments compare their presences in mangrove and to highlight the specific roles in carbon biogeochemical cycling. That would improve the quality of the MS. Therefore, we follows the suggestions and modified the discussion part (Line 307-318).

Q7: One of the important groups that was reported from this mangrove is the presence and dominance of Firmicutes. However, I do not see any discussion on this point given Firmicutes signal were found to be much higher in Sundarbans compared to other mangroves.

Answer: The members from Firmicutes not much dominated in these study sites. As mentioned the manuscript the top ten dominated bacterial phylum are Proteobacteria, Bacteroidetes, Actinobacteria, Planctomycetes, Verrucomicrobia, Cyanobacteria, Acidobacteria, Chloroflexi, Gemmatimonadetes, and Nitrospirae. 

Q8: It is really interesting to see the relative dominance of Thaumarchaeota Marine Group (MG I) and Euryarchaea MG II in this ecosystem. I am just curious if the authors can link this finding with reported dissolved ammonium concentration in this sector of Sundarbans given the availability of ammonia seems to be episodic in nature and thus there are consequences for nitrification rates. Also, the selection of primers improved the resolution in terms of encountering these groups? It is also important to highlight given the sampling has been focused only one time point, the observed community structure may also exhibit temporal variability.

Answer: Thank you for asking this question. This issue would be an interesting perspective to investigate with reference to current data. In fact in the subsequent project we are proposing to investigate this objective. With the current data set, I am afraid we can really do that, as we have not measured the concentration of dissolved ammonium during the investigation on 2017. But we have framed one collaborative project to investigate the nitrification rate and responsible microbial community structure in Sundarban Mangrove.

---

## [Decision Letter · Decision Letter 1]

21 Jan 2020

PONE-D-19-22253R1

Insights on aquatic microbiome of the Indian Sundarbans mangrove areas

PLOS ONE

Dear Dr. Dhal,

Thank you for submitting your manuscript to PLOS ONE. After careful consideration, we feel that it has merit but does not fully meet PLOS ONE’s publication criteria as it currently stands. Therefore, we invite you to submit a revised version of the manuscript that addresses the points raised during the review process.

We would appreciate receiving your revised manuscript by Mar 06 2020 11:59PM. To enhance the reproducibility of your results, we recommend that if applicable you deposit your laboratory protocols in protocols.io, where a protocol can be assigned its own identifier (DOI) such that it can be cited independently in the future. For instructions see: http://journals.plos.org/plosone/s/submission-guidelines#loc-laboratory-protocols

We look forward to receiving your revised manuscript.

Kind regards,

Jiang-Shiou Hwang, Ph.D.

Academic Editor

PLOS ONE

Reviewers' comments:

Reviewer's Responses to Questions

**Comments to the Author**

1. If the authors have adequately addressed your comments raised in a previous round of review and you feel that this manuscript is now acceptable for publication, you may indicate that here to bypass the “Comments to the Author” section, enter your conflict of interest statement in the “Confidential to Editor” section, and submit your "Accept" recommendation.

Reviewer #1: All comments have been addressed

Reviewer #2: All comments have been addressed

2. Is the manuscript technically sound, and do the data support the conclusions?

Reviewer #1: Yes

Reviewer #2: Yes

3. Has the statistical analysis been performed appropriately and rigorously? 

Reviewer #1: Yes

Reviewer #2: Yes

4. Have the authors made all data underlying the findings in their manuscript fully available?

Reviewer #1: No

Reviewer #2: Yes

5. Is the manuscript presented in an intelligible fashion and written in standard English?

Reviewer #1: Yes

Reviewer #2: Yes

6. Review Comments to the Author

Reviewer #1: PLOS ONE

Manuscript: PONE-D-19-22253-R1

The authors have revised and improved the manuscript. Several parameters are concerned. Suggest conduct minor modification.

1. Lines 121-133: It said physicochemical parameters (salinity and pH), DOC, nutrient, nitrate, nitrite, combined nitrite and nitrate etc. However, they did not report measurements of TN and DIN (Table 1).

2. Line 131-132: It said nitrite, nitrate, phosphate, silicate were analyzed using continuous flow analyzer. Suggest report clearly the measurements.

3. Line 132: It said ‘phosphate (PO43-)”. In Table 1, it reported TN (total nitrogen), DIN (dissolved inorganic nitrogen), and PO4. Nitrogen and phosphorus are important nutrients. Suggest check the measurement is orthophosphate (PO43-), not including pyrophosphate and metaphosphate. It is fine to use phosphate (PO43-). The orthoposphate (PO43-) is commonly used and reported as phosphate (PO43-). However, suggest check the standard solution, and check the measurement is phosphorus or phosphate. Suggest change the term here to orthophosphate-P or phosphate-P (PO43--P) to match well with DIN. Suggest check the term PO4 in Table 1. It is not correct. Suggest use orthophosphate-P, or simply use PO43--P.

4. Line 132: It said silicate (Si). In Table 1, it said Si. They are not correct. The term of Si is silicon, not silicate. Readers know silicate is important, and the measurement of silica (SiO2) is used for evaluating content of silicate. Suggest check the standard solution and measurement of this parameter. I think it is silica (SiO2), not Si, not silicate. Please check it is silica (SiO2) or silicon (Si). Suggest report silica, not silicate, not silicon.

5. Line 125 and Table 1: It said DOC and said DOC (µM). Suggest report clearly the measurement. Suggest check the unit “µM”. Suggest check the standard and report which chemical is used as standard.

Reviewer #2: The authors have addressed the points that I raised earlier. The manuscript is now acceptable for publication.

7. PLOS authors have the option to publish the peer review history of their article (what does this mean?). If published, this will include your full peer review and any attached files.

Reviewer #1: No

Reviewer #2: Yes: Punyasloke Bhadury

---

## [Author Response · Author response to Decision Letter 1]

30 Jan 2020

Review Comments to the Author

Reviewer #1: 

P1. Lines 121-133: It said physicochemical parameters (salinity and pH), DOC, nutrient, nitrate, nitrite, combined nitrite and nitrate etc. However, they did not report measurements of TN and DIN (Table 1).

Answer: Yes. Its mistake. As I mentioned earlier, this work has been carried out in ZMT, Germany under expert supervision. The total nitrogen (TN) content of each the samples were measured using the EuroVector EA 3000 elemental analyzer. The DIN measurements were done based on the methods described by Grasshoff et al. 1999. (Page 6, line 127-128)

P2. Line 131-132: It said nitrite, nitrate, phosphate, silicate were analyzed using continuous flow analyzer. Suggest report clearly the measurements.

Answer: Thanks for this rising this issue and we also agree with the reviewer suggestions on including the details of this information. In this connection, this is to bring your notice that, the Dissolved inorganic nutrients that includes combined nitrate and nitrite (NOx), phosphate (PO43--P), and silicate Si(OH)4-) were analyzed using a continuous flow analyzer (Flowsys by Unity Scientific, Brookfield, USA) and detected spectrophotometrically as a colored complex (Grasshoff et al. 1999). To correct for the water color, absorption of the samples was measured without chemicals first, and the values were subtracted from those obtained with chemicals. Precision of the method was better and most of the nutrient concentrations were near the detection limit.

P3. Line 132: Line 132: It said ‘phosphate (PO43-)”. In Table 1, it reported TN (total nitrogen), DIN (dissolved inorganic nitrogen), and PO4. Nitrogen and phosphorus are important nutrients. Suggest check the measurement is orthophosphate (PO43-), not including pyrophosphate and metaphosphate. It is fine to use phosphate (PO43-). The orthoposphate (PO43-) is commonly used and reported as phosphate (PO43-). However, suggest check the standard solution, and check the measurement is phosphorus or phosphate. Suggest change the term here to orthophosphate-P or phosphate-P (PO43--P) to match well with DIN. Suggest check the term PO4 in Table 1. It is not correct. Suggest use orthophosphate-P, or simply use PO43--P

Answer: Thanks for suggestions but we are sorry for the confusion rises again. We checked with the methods again as I had little knowledge with such details. This part of comments is enriching. After getting detail information it is now clear that its actually orthophosphate (PO43-) that we measured and now in the current manuscript for both text and Table 1, this information is updated as PO43--P (Lines 262).

P4. Line 132: It said silicate (Si). In Table 1, it said Si. They are not correct. The term of Si is silicon, not silicate. Readers know silicate is important, and the measurement of silica (SiO2) is used for evaluating content of silicate. Suggest check the standard solution and measurement of this parameter. I think it is silica (SiO2), not Si, not silicate. Please check it is silica (SiO2) or silicon (Si). Suggest report silica, not silicate, not silicon.

Answer: Yes. Again thanks for raising this issue. Actually it’s a silicate (Si(OH)4-) that we measured. For the calibration we used Sodium hexafluorosilicate (Na2SiF6) during measurements.

 P5. 5. Line 125 and Table 1: It said DOC and said DOC (µM). Suggest report clearly the measurement. Suggest check the unit “µM”. Suggest check the standard and report which chemical is used as standard.

Answer: Yes. The DOC is measured in µM. The DOC were measured by the high-temperature catalytic oxidation method on a Shimadzu TOC-VCPH total organic carbon analyser. The instrument was calibrated with a 10-point calibration curve of serial dilutions from a potassium hydrogen phthalate certified stock solution (1000 ppm Standard Fluka 76067-500ML-F). Consensus reference material provided by DA Hansell and W Chan of the University of Miami was used a positive control between every 10 samples. Each sample was measured with five replicate injections.

---

## [Editor Report · Decision Letter 2]

5 Feb 2020

Insights on aquatic microbiome of the Indian Sundarbans mangrove areas

PONE-D-19-22253R2

Dear Dr. Dhal,

We are pleased to inform you that your manuscript has been judged scientifically suitable for publication and will be formally accepted for publication once it complies with all outstanding technical requirements.

With kind regards,

Jiang-Shiou Hwang, Ph.D.

Academic Editor

PLOS ONE
---

## [Editor Report · Acceptance letter]

10 Feb 2020

PONE-D-19-22253R2 

Insights on aquatic microbiome of the Indian Sundarbans mangrove areas 

Dear Dr. Dhal:

I am pleased to inform you that your manuscript has been deemed suitable for publication in PLOS ONE. Congratulations! Your manuscript is now with our production department. 

With kind regards,

on behalf of

Prof. Jiang-Shiou Hwang 

Academic Editor

PLOS ONE